# Quantification of 363 Pesticides in Leafy Vegetables (Dill, Rocket and Parsley) in the Turkey Market by Using QuEChERS with LC-MS/MS and GC-MS/MS

**DOI:** 10.3390/foods12051034

**Published:** 2023-02-28

**Authors:** Bilge Deveci, Ozgur Golge, Bulent Kabak

**Affiliations:** 1Department of Food Engineering, Faculty of Engineering, Hitit University, Corum 19030, Turkey; 2Department of Gastronomy and Culinary Arts, Faculty of Tourism, Alanya Alaaddin Keykubat University, Alanya 07425, Turkey

**Keywords:** food safety, green leafy vegetables, GC-MS/MS, LC-MS/MS, method validation, pesticide residues

## Abstract

Contamination of agricultural products with pesticide residues is a growing concern due to their adverse health effects and increasing worldwide usage of pesticides. In 2021 a total of 200 samples of green leafy vegetables, including 80 dill, 80 rocket and 40 parsley, purchased from greengrocer shops, markets and bazaars in Corum Province, Turkey, were monitored for pesticide residues. In green leafy vegetables, 363 pesticides were analyzed using a quick, easy, cheap, effective, rugged, and safe (QuEChERS) sample preparation, followed by liquid chromatography coupled to mass spectrometry (LC-MS/MS) for 311 residues and gas chromatography coupled to mass spectrometry (GC-MS/MS) for 52 residues. The method was in-house validated at two fortification levels, and satisfactory recoveries and precisions were achieved for all residues. No quantifiable residues were found in 35% of the samples, whereas 43 residues belonging to 24 different chemical classes were detected in 130 green leafy vegetables. Among the green leafy vegetables, the highest occurrence frequency was recorded in the rocket, followed by dill and parsley. In 46% of the green leafy vegetables, the residue levels exceeded European Union Maximum Residue Levels (EU MRLs). The most frequently detected pesticides were pendimethalin (22.5%), diuron (38.7%) and pymetrozine (52.5%) in dill, rocket and parsley, respectively.

## 1. Introduction

Turkey has extensive resources of land and water and rare agroecological conditions. The total utilized agricultural land, including permanent meadows and pastures, is over 37 million ha, of which 41.4% consists of the area of crop products and cereals [1]. The vegetable production of Turkey reached over 31.6 million metric tonnes in 2022. Green leafy vegetable production in Turkey was more than 2.2 million metric tonnes in the 2021 season, with 39% cabbage, 24.5% lettuce, 10.7% cauliflower, 9.9% spinach, 4.9% parsley, 4.8% broccoli, 1.2% rocket, 0.4% dill and 4.6% others. Green leafy vegetables, including parsley, rocket and dill, are primarily cultivated in the Mediterranean region of Turkey, particularly in Hatay Province, followed by Thrace and Aegean regions [2]. Fresh fruits are marketed by passing through many different routes from the point of production. The distribution channels in green leafy vegetables are producer, wholesaler, wholesale market, retailer and consumer.

Green leafy vegetables such as dill, parsley, rocket, lettuce and spinach, an important part of the healthy diet, are among the most popular locally grown vegetables today. Green leafy vegetables are rich in dietary fiber, minerals (Mg, K, iron), phytochemicals (β-carotene, lutein) and vitamins, particularly folate, vitamins A and C and fat-soluble vitamin K [3,4]. These are often consumed fresh as a salad as well as can be cooked or dried and stored for use throughout the year [3]. In recent years, the demand for fresh green leafy vegetables has been increasing due to the changing consumption habits for a healthy life [5].

On the other hand, these products, which are grown close to the ground, can be an important source of exposure to pathogenic bacteria such as Salmonella and *Escherichia coli*, depending on their raw consumption [6]. Moreover, there is an increasing awareness of their chemical food safety hazards such as heavy metals and pesticides. Pesticides are mostly chemical-based plant production products and are used to protect the plant from damage by various diseases, insects, fungi, rodents and weeds [7]. In 2020, 2.66 million metric tonnes of pesticides were used in agriculture, which accounted for 1.81 kg/ha of pesticides per cropland area. Turkey was the third largest pesticide consumer country in Europe, with 53,672 metric tonnes of pesticide applications (2% of global consumption) for agricultural use (which accounted for 2.32 kg/ha pesticides per cropland area), after France and Italy. Fungicides were the most used class of pesticides in Turkey, with 20,600 metric tonnes, followed by insecticides and herbicides, whereas herbicides were the top-used class of pesticides in the world [8].

Pesticides can prevent losses in agricultural products caused by pests, increase product quality and yield, ensure the food supply and extend the storage period of the product. However, pesticide residues resulting from the use of plant protection products on food are a growing concern because of their adverse acute and chronic health effects and environmental problems. Long-term exposure to pesticides has been associated with a range of health problems, including cardiovascular diseases, neurological and developmental toxicity, genetic changes, birth defects and cancer [9]. For this reason, pesticide residues need to be remediated to prevent adverse effects on humans and animals. The use of microorganisms such as microalgae is the most effective, eco-friendly and economical method in bioremediation to detoxify or convert hazardous compounds into non-toxic compounds [10].

Different sample preparation and analytical methods have been used for monitoring pesticide residues in various food products. Sample preparation techniques used for pesticide analysis include liquid-liquid extraction [11], liquid-phase microextraction [12], solid-phase extraction [13,14], solid-phase microextraction [15], accelerated solvent extraction [16], supercritical fluid extraction [17,18], matrix solid-phase dispersion [19,20] and microwave-assisted extraction [21]. However, the majority of these techniques are rather time-consuming, labor-intensive, complicated and expensive and produce considerable quantities of waste. Alternatively, the QuEChERS (quick, easy, cheap, effective, rugged and safe) method developed by Anastassiades et al. [22] has gained significant popularity as the method of choice for agricultural and animal-based matrices within the last two decades due to its simplicity, speed, low cost, high throughput and minimal solvent requirement. 

Non-chromatographic techniques such as enzyme-linked immunosorbent assays (ELISA) [23] or bioassays [24] are of minor relevance in pesticide analysis. Liquid chromatography (LC) and gas chromatography (GC) coupled with triple quadrupole mass detectors (MS) are the most powerful techniques and have been used for the accurate simultaneous determination of multiclass pesticide residues in agricultural matrices [25,26,27]. With these techniques, limits of quantification (LOQs) below 0.01 mg kg^−1^ for pesticide residues can be reached. While the residues of sulfonyl or benzoyl ureas and many carbamates or triazines are analyzed by LC-methods, the residues belonging to the class of organophosphorus, organochlorines, pyrethroids and nitrogen-containing compounds can be better separated by GC methods [28,29].

In general, vegetables are less prone to insects and diseases when compared to fruits, so pesticide residues in vegetables are lower. However, several studies have shown that multiple residues of pesticides can be found in high levels on leafy vegetables [30,31,32,33]. The consumption of leafy vegetables with high levels of active substances may decrease the beneficial effects of their consumption. Monitoring of residues in ready-to-eat foods such as fruits and leafy vegetables through risk assessment and regulation of pesticide use in agricultural production plays an important role in public health, national security and trade. Thus, the main aim of the present study was to determine the occurrence and concentration of multiclass pesticide residues in dill (*Anethum graveolens*), rocket (*Eruca vesicaria*) and parsley (*Petroselinum crispum*) available in the Turkish market and to promote discussion about the importance of monitoring pesticides in leafy vegetables. To achieve this goal, a modified version of QuEChERS procedure, followed by liquid chromatography-tandem mass spectrometry (LC-MS/MS) and gas chromatography-tandem mass spectrometry (GC-MS/MS) analysis, was used.

## 2. Materials and Methods

### 2.1. Samples

A total of 200 fresh green leafy vegetable samples consisting of dill (*n*= 80), rocket (*n*= 80) and parsley (*n* = 40) were randomly collected from supermarkets, groceries and bazaars in Corum, Turkey, between May and July 2021. The samples were taken from seven different supermarkets, eight different groceries and five different bazaars in Corum Province. While each supermarket uses its own distribution channels, six green leafy vegetable suppliers are available for all groceries and bazaars in Corum. Two bundles of each of the three types of green leafy vegetable samples (approximately 100 g) were selected and removed from their non-renewable parts. The samples were chopped, homogenized with a home food processor (Phillips HR 7770, Royal Philips Electronics NV, Istanbul, Turkey) and stored in a plastic container in a refrigerator (4–8 °C) until analysis.

### 2.2. Reagents, Chemicals and Standards

Analytical grades of anhydrous magnesium sulphate (MgSO_4_) and sodium acetate (CH_3_COONa), LC-MS grade formic acid (98% purity) and charcoal activated (CA) powder were supplied from Merck (Darmstadt, Germany). Acetonitrile (MeCN) and methanol (MeOH) with chromatography grade glacial acetic acid (CH_3_COOH) and analytical grade ammonium formate (HCOONH_4_, ≥99% purity) were ordered from Sigma–Aldrich (Steinheim, Germany). Primary-secondary amine (PSA, 40 µm particle size) was obtained from Supelco (Bellefonte, PA, USA). Ultrapure water for sample preparation and chromatographic analysis was obtained in the laboratory using a Milli Q water purification system (Millipore, Molsheim, France). A total of 363 standards of pesticides (311 LC amenable and 52 GC amenable) were available for the present study. The selected active substances included the commonly used pesticides at different stages in the cultivation of green leafy vegetables, as well as non-registered and banned pesticides. The standards were acquired from four suppliers, A2S Analytical Standard Solutions Co. (Saint Jean d’Illac, France), ChemService (West Chester, PA, USA), Dr. Ehrenstorfer GmbH (Augsburg, Germany) and Sigma-Aldrich (Steinheim, Germany). The purities of standards were ≥95%. Stock solutions of pesticide mixtures were prepared at 10 mg L^−1^ in acetonitrile and stored at −18 °C in glass vials. Multiresidue working solutions were prepared daily and used for the preparation of matrix-matched calibration standards and validation study.

### 2.3. Sample Preparation

Pesticide residues were extracted from green leafy vegetables using the QuEChERS sample preparation technique according to the AOAC official method [34], with minor modifications. The workflow of the sample preparation step is illustrated in Figure 1. Briefly, 7.5 g of the chopped green leafy vegetable sample was weighed into a 50 mL extraction tube, and then 15 mL of acetonitrile (containing 1% acetic acid) was added. The tube was rotated (Multi RS-60 rotator, Biosan, Riga, Letonya) vigorously for 2 min and vortexed for 1 min on a vortex mixer (Heidolph, Schwabach, Germany). Then, 6 g of MgSO_4_ and 1.5 g of anhydrous sodium acetate were added, and the tube was rotated for 2 min and vortexed for 1 min. The tube was centrifuged at 5000 rpm for 5 min at room temperature. Afterwards, 2 mL of the supernatant was transferred to a 15 mL centrifuge tube containing 300 mg of MgSO_4_, 100 mg of PSA and 25 mg of CA. The mixture was rotated vigorously, vortexed and then centrifuged at 5000 rpm for 3 min at room temperature. After centrifugation, the supernatant (0.5 mL) was diluted with water (1:3 ratios, *v*/*v*), and the diluted extract was transferred into the LC vial for chromatographic analysis.

### 2.4. LC-MS/MS Analysis

Separation of 311 LC-amenable pesticide residues in green leafy vegetable matrices was carried out using the ThermoFisher Scientific UltiMate 3000 HPLC system (Bremen, Germany). This was equipped with an autosampler, binary pump, degasser and column oven. The chromatographic conditions are summarized in Table 1. The mass system included a ThermoFisher Scientific Q-Exactive Focus Orbitrap MS (Bremen, Germany) with an electrospray ionization (ESI) interface. The spray and vaporizer temperatures were set at 320 and 295 °C, respectively. The spray voltage was 2.8 kV. Data acquisition and data processing were performed with Thermo Scientific Xcalibur v.4.0 software. Two specific multiple reaction monitoring (MRM) transitions were monitored for each pesticide, and both pairs of the MRM transitions were used for confirmation analysis according to SANTE/11312/2021 Guidelines [35]. The first transition, which corresponds to the most abundant product ion, was used for quantification analysis. The MRM transitions used for screening and quantification of 311 pesticides are shown in Appendix A.

### 2.5. GC-MS/MS Analysis

For the analysis of 52 GC-amenable pesticides in green leafy vegetables, GC-MS/MS (GC-MS TQ8050, Shimadzu, Tokyo, Japan) with multiple reaction monitoring (MRM) acquisition mode was used. The column used for the chromatographic separation of analytes was an HP-5MS (30 m × 0.25 mm × 0.25 µm, Agilent Technologies, Palo Alto, CA, USA). High-purity helium at a constant flow rate of 2.4 mL min^−1^ was used as the carrier gas. Two microliters of the extract were injected in splitless mode. The column oven temperature was programmed at 50 °C, held for 0.0 min, ramped at 50 °C/min to 150°C, ramped at 20 °C/min to 230 °C and held for 1 min, and finally ramped at 8 °C/min to 290 °C, at which it was held for 18.5 min. The MS transfer line and ion source temperatures were sat at 280 and 200 °C, respectively. The MS operated in the electron impact ionization (EI) at 70 eV. Two ion transitions at the experimentally optimized collision energy were monitored for each target compound (Appendix A). Both pairs of the MRM transitions were used for confirmation analysis, and the most sensitive transitions were selected for quantification analysis.

### 2.6. Method Validation

The analytical method was validated in-house according to SANTE/11312/2021 Guidelines [35]. To study linearity, matrix-matched calibration curves were created by spiking the pesticide mix from 0.005 to 0.100 mg kg^−1^ in blank rocket extract as a representative matrix. To evaluate the limit of quantification (LOQ), recovery and precision parameters, studies were carried out on spiked rocket samples. LOQ was calculated via the standard deviation of blank rocket matrices spiked at a low level of pesticide mix that can be quantified with acceptable accuracy (70–120%) and precision (relative standard deviations (RSD) ≤20%) as described in the SANTE/11312/2021 guidelines. To assess the method recovery and precision, two sets of the five blank rocket samples were spiked with all pesticides at two different concentrations (the targeted limit of quantification of 0.01 mg kg^−1^ and 5× targeted LOQ (0.05 mg kg^−1^)), extracted according to sample preparation protocol and quantified against the matrix-matched calibration curves. The repeatability was determined on the same day by one operator (*n* = 5), while within-laboratory reproducibility was assessed on five consecutive days by two different operators (*n* = 10). The expanded measurement uncertainty (⋃′) was measured by multiplying combined uncertainty associated with within-laboratory reproducibility and trueness (bias), with a coverage factor of *k* = 2, as described in detail previously [36].

## 3. Results

### 3.1. Validation Data

The treated samples were analyzed in MRM acquisition mode using LC-MS/MS for 311 pesticide residues and GC-MS/MS for 52 residues. The target residues were identified with the transition ions at the retention time and ion ratio data corresponding to those of the reference standards. The ion ratios were within the ±30% as described by SANTE/11312/2021 guideline.

The linearity of the calibration curve for each active substance was evaluated by generating a five-point calibration curve with the range of 0.005–0.100 mg kg^−1^. In the green leafy vegetable extract, linearity showed satisfactory results (coefficient of determination (*R^2^*) greater than 0.99) for all residues. The results of in-house validation data (LOQ, recovery, precision and uncertainty) across the target LC-amenable and GC-amenable analytes in green leafy vegetable matrices are summarized in Appendix A, respectively. As shown in Appendix A, the LOQs of LC-amenable and GC-amenable pesticides were within the range of 0.003–0.011 mg kg^−1^ and 0.008–0.013 mg kg^−1^, respectively.

The trueness, expressed by measured recovery, was calculated using the data from the analysis of spiked samples at two different concentrations. The mean recoveries for each set of five spikes, at two different concentrations (0.01 and 0.05 mg kg^−1^) prepared and analyzed over five days by two different analysts, were within the range 73 to 116% and hence were well within the range of performance criteria in the SANTE guidelines. The repeatability (1.4–19.8%) and reproducibility (3.4–20.0%) of the method were also excellent for all compounds at two different concentrations in rocket extracts. The expanded measurement uncertainties were within the range of 19–50%, the values of which are compliant with provisions set in SANTE guidelines.

### 3.2. Results on Pesticide Residues in Green Leafy Vegetables

In 2021 a total of 200 samples of green leafy vegetables from conventional cultivation were analyzed for 363 pesticide residues. In green leafy vegetables, 43 different residues were detected with a rate of 0.214 mg pesticide per kg, which means the detected substances are often found in only small concentrations. There were exceedances of the MRL in 92 (46%) leafy vegetable samples. Residues exceeding the MRL were related to 29 different pesticides. The groups of pesticides based on chemical class detected in green leafy vegetables are shown in Figure 2.

In green leafy vegetable samples, 24 different groups of pesticides based on chemical class were detected. It is not possible to draw definitive conclusions on which chemical class of pesticides is mainly absorbed into the leafy vegetables from the present study, due to the restricted sample size. However, the class of triazole (difenoconazole, hexaconazole, penconazole, tebuconazole, triflumizole and triadimenol) had the highest mean concentration (1.204 mg kg^−1^) in dill samples, followed by strobilurin (1.092 mg kg^−1^), organophosphate (1.051 mg kg^−1^) and benzamides (0.78 mg kg^−1^). Organophosphate (1.908 mg kg^−1^) and triazole (1.708 mg kg^−1^) were also the first two groups of pesticides detected in parsley with higher mean concentrations. Triazine (pymetrozine) was detected with the third highest mean concentration of any pesticide group in parsley, whereas it was not detected in dill and rocket samples. In rocket samples, oxadiazine had the highest mean concentration (1.615 mg kg^−1^), followed by carbamate (0.55 mg kg^−1^), neonicotinoid (0.472 mg kg^−1^) and pyrethroid (0.374 mg kg^−1^).

#### 3.2.1. Residue Data on Dill

The distribution of pesticides measured in dill samples is presented in Table 2. In 42.5% of the dill samples, no quantifiable residues were recorded, while 46 samples (67.5%) contained measurable residues at or above the LOQ. Thirty of the eighty dill samples contained residues in concentrations higher than EU MRLs. The MRL was exceeded for 17 different pesticides.

Only one pesticide was quantified in 21 samples, whereas more than one pesticide (up to 15 residues) substance was quantified in 25 samples (Figure 3.). In total, 32 different pesticides were found in dill samples in concentrations higher than LOQ, including fourteen fungicides, twelve insecticides and six herbicides. Twelve of these substances are non-approved residues in green leafy vegetables in the EU, while the rest (20 residues) are currently approved in the EU.

Among the residues, the most frequently recorded pesticides quantified in more than 10% of the dill samples were pendimethalin (22.5%), tebuconazole (13.7%) and fluopyram (11.5%), with concentrations ranging from 0.016 to 0.077 mg kg^−1^ (mean = 0.037 mg kg^−1^), 0.011 to 2.077 mg kg^−1^ (0.523 mg kg^−1^) and 0.016 to 3.915 mg kg^−1^ (mean = 0.780 mg kg^−1^), respectively. Both pendimethalin and tebuconazole were measured in four dill samples in concentrations higher than EU MRLs of 0.05 and 1.5 mg kg^−1^, respectively. The 29 other residues were detected in less than 10% of the samples, with an incidence rate varying from 1.2% to 7.5%.

#### 3.2.2. Residue Data on Rocket

The distribution of pesticides measured in rocket samples is shown in Table 3. Compared to dill, higher occurrence frequencies were recorded in rocket samples. While 23.7% of the rocket samples were free from the residues analyzed, 61 rocket samples contained at least one residue in quantifiable concentrations. In 35% of the rocket samples, only one residue was recorded. The remaining 33 rocket samples (41.3%) contained more than one residue (Figure 4), the frequency of which was higher than the result of dill samples (31.3%). The concentrations of residues in 41 samples were higher than EU MRLs.

A total of 23 substances were measured in rocket samples: 12 insecticides, 9 fungicides, 1 herbicide and 1 acaricide. While 15 of the 23 residues detected in the rocket samples are approved by the EU, 8 of them are not approved for use. The herbicide diuron was the predominant residue in rocket samples, with a frequency rate of 38.7%. It was measured in concentrations of 0.011–0.042 mg kg^−1^, with a mean level of 0.036 mg kg^−1^. Diuron is no longer approved as an active substance in the EU after September 2020 due to concerns over its toxicity and the threat posed to people. Acetamiprid was the second most frequently detected residue in rocket samples, with a 30% frequency of detection. The concentrations of acetamiprid in 24 rocket samples varied from 0.010 to 0.471 mg kg^−1^ (mean = 0.180 mg kg^−1^), which were below the EU MRL of 3 mg kg^−1^. Acetamiprid is approved as a broad-spectrum, systemic and contact-action neonicotinoid insecticide until the date 2033. In rocket samples, the third most frequently recorded substance was deltamethrin, with a frequency rate of 10%. The insecticide deltamethrin was measured in eight rocket samples and in low concentrations (0.013–0.07 mg kg^−1^). The 20 other pesticides quantified in less than 10% of the samples analyzed were thiamethoxam (8.7%), metalaxyl-M (7.5%), azoxystrobin (6.3%), cypermethrin (6.3%), lambda-cyhalothrin (6.3%), dinocap (5%), fluopyram (3.8%), indoxacarb (3.7%), pyridaben (3.7%), bifenthrin (2.5%), clothianidin (2.5%), dimethomorph (1.2%), hexythiazox (1.2%), imidacloprid (1.2%), oxamyl (1.2%), penconazole (1.2%), pirimicarb (1.2%), tebuconazole (1.2%), triadimenol (1.2%) and trifloxystrobin (1.2%).

#### 3.2.3. Residue Data on Parsley

The distribution of pesticides measured in parsley samples is shown in Table 4. The frequency of quantifiable results in parsley samples was 57.5%, whereas 17 parsley samples were free from target substances. In 13 parsley samples, only one pesticide was measured. Residues of more than one pesticide were recorded in 25% of parsley samples; up to eight different compounds were observed in individual parsley samples (Figure 5).

The parsley samples had residues of 14 different pesticides in 23 samples; 21 of them contained residues above the EU MRL. Half of the active substances measured in parsley samples are non-approved residues in the EU, while the other half (seven substances) are currently approved in the EU. Insecticide was the most frequently detected pesticide type in parsley samples with eight substances, followed by fungicide (five substances) and herbicide (one substance).

Amongst the target substances, pymetrozine was the most frequently detected (52.5%) residue in parsley samples, followed by acetamiprid (12.5%) and imidacloprid (10%). The non-approved pymetrozine insecticide was measured in 21 parsley samples in concentrations of 0.016–2.437 mg kg^−1^, with a mean level of 1.351 mg kg^−1^. In addition, in 20 parsley samples, the residue levels exceeded the established EU MRL of concern (0.05 mg kg^−1^). The other insecticide, acetamiprid, was found to have a relatively low occurrence frequency in parsley samples in concentrations of 0.014–0.074 mg kg^−1^ (mean = 0.044 mg kg^−1^), also below the EU tolerance level. Non-approved imidacloprid residue was recorded in four samples, with levels of 0.012–0.040 mg kg^−1^, which were below the EU MRL of 0.05 mg kg^−1^. The pesticides with a lower frequency of detections in parsley samples were cypermethrin (7.5%), fluopyram (5%), pendimethalin (5%), chlorpyrifos-methyl (2.5%), ethoprophos (2.5%), hexaconazole (2.5%), indoxacarb (2.5%), malathion (2.5%), methiocarb (2.5%), pyrimethanil (2.5%) and tebuconazole (2.5%). Among target compounds, only one herbicide, pendimethalin, was measured in two parsley samples in concentrations of 0.013–0.020 mg kg^−1^.

While acetamiprid, azoxystrobin, deltamethrin and quizalofop-p are the most common pesticides used by farmers in the cultivation of conventional green leafy vegetables in Turkey, many other residues can also be detected in the product. The presence of many different pesticides in leafy vegetables could be not only through the direct application of active substances to leafy vegetables but also through other sources such as water resources and the environment. Pesticides can be transported by factors such as wind erosion, evaporation and attachment to soil particles. The main factors affecting the transport are water solubility (hydrophobic/hydrophilic), volatility, adsorption in water or soil, bioaccumulation amount in soil and biota, and chemical/microbial degradation properties [37]. Several studies have been performed to detect pesticide contamination in surface water and sediments in Turkey. In a previous study, heptachlor epoxide had the highest mean concentration (0.28 ng L^−1^) among organochlorine pesticides in the surface water of the Küçük Menderes River in Turkey [38]. More recently, Tokatlı [39] monitored 174 pesticides in the Ergene River Basin in Turkey. The number of pesticides reported in surface water and sediment samples was 19 and 26, respectively. Carbendazim and azoxystrobin were the most frequently detected pesticides in surface water. For sediment, the predominant pesticides were prochloraz, tebuconazole and azoxystrobin. In another study, the levels of 16 organochlorine pesticides, which are more resistant to environmental degradation, in surface water in the mid-Black Sea region, Turkey, were monitored. Among organochlorine pesticides, hexachlorocyclohexane, dieldrin, p,p-DDD and heptachlor epoxide were the most commonly detected residues [40]. During a one-year study, eight pesticides (malathion, etofenprox, molinate, oxamyl, propamocarb, tebufenozide, linuron and piperonyl butoxide) were detected in surface water and sediment in Karaboğaz Lake, Northern Turkey [41].

Concerning the pesticide residues in green leafy vegetables in Turkey, only a few studies were available. It was shown by Esturk et al. [42] that parsley samples (*n* = 40) contained 14 different residues, and seven or more active substances were detected in each parsley sample. The pesticides carbendazim, cymoxanil, cypermethrin and dichlorvos were determined in all parsley samples. More recently, Balkan and Yılmaz [32] analyzed 74 green leafy vegetable samples from Turkey for the monitoring of 260 pesticide residues. Only thirteen different active substances were recorded in 57.6% of the samples, with acetamiprid, cypermethrin, deltamethrin and pyraclostrobin being the most frequently quantified compounds. 

In contrast to our findings, imidacloprid, chlorpyrifos, profenofos and cypermethrin were the most frequently detected residues in leafy vegetables from Sagar, India [33]. In South Korea, a total of 17,977 leafy vegetables were monitored for 230 kinds of pesticide residues over a period of 15 years. The results showed that 15.7% of the samples contained at least one pesticide, and the samples exceeding the MRLs were mostly noted in spinach, ssamchoo, crown daisy, lettuce and perilla leaves. Azoxystrobin, dimethomorph, procymidone, indoxacarb, lufenuron and endosulfan were the predominant residues in leafy vegetables [43]. In another study in Korea, a total of 34,520 vegetable samples were analyzed for 283 pesticides, and the frequency of pesticides was found to be 19.2% in leafy vegetables. The results observed that 4 out of 78 rocket samples and 33 out of 87 parsley samples were found to contain at least one residue [44]. In a monitoring program conducted by the Danish Veterinary and Food Administration between 2004–2011, a total of 371 lettuce samples grown in Denmark or nine other countries were checked for about 250 pesticides. Of the lettuce samples originating from Denmark, 13% contained 8 residues, whereas the samples of foreign origins had 36 different residues in 45% of the samples [45].

In 2014–2015 a total of 36 pesticides were monitored in 118 green leafy vegetables in Northern Chile, and they were observed in 84.7% of the samples. In 27% of the samples, the levels exceeded EU MRLs of concern. The most abundant pesticides in leafy vegetables were boscalid, chlorpyrifos, lambda-cyhalothrin and methamidophos [30]. In Italy, 300 samples of green leafy vegetables were monitored for 210 active substances, and 53.3% of the samples contained at least one residue. The EU MRLs were exceeded for only paclobutrazol and tau-fluvalinate in multiresidual samples [31]. In the 2019 European Union report on pesticide residues in food, 11% of 245 rocket samples contained only one residue, while multiple residues were quantified in 67.7% of the rocket samples. For parsley samples, pesticide residues were reported in 68.2% of 277 samples, 138 of which contained multiple residues [46].

## 4. Conclusions

A method based on the QuEChERS sample preparation procedure and LC-MS/MS or GC-MS/MS was validated for residual analysis of 363 pesticides in green leafy vegetables. The method provided good results for recovery rate, precision, sensitivity and linearity. It was demonstrated that the use of both PSA and charcoal activated in the QuEChERS clean-up step allows the removal of sugars, organic acids, fatty acids, chlorophyll and other non-polar pigments prior to LC-MS/MS and GC-MS/MS analysis, and the analytical method is useful for the analysis of multiclass residues in green leafy vegetables. The in-house validated method was applied to 200 green leafy vegetables, including 80 dill, 80 rocket and 40 parsley samples from Turkey. Of the green leafy vegetable samples studied, 65% contained at least one quantifiable residue. Ninety-two of these samples exceeded the EU MRL of concern. Pendimethalin, diuron and pymetrozine were the most frequently detected residues in dill, rocket and parsley, respectively. To protect consumer health, the use of plant protection products must be routinely monitored and controlled by the General Directorate of Food and Control of the Ministry of Agriculture and Forestry. Moreover, sustainable remediation strategies for agricultural pollutants should be studied to eradicate their adverse effects.

## Figures and Tables

**Figure 1 foods-12-01034-f001:**
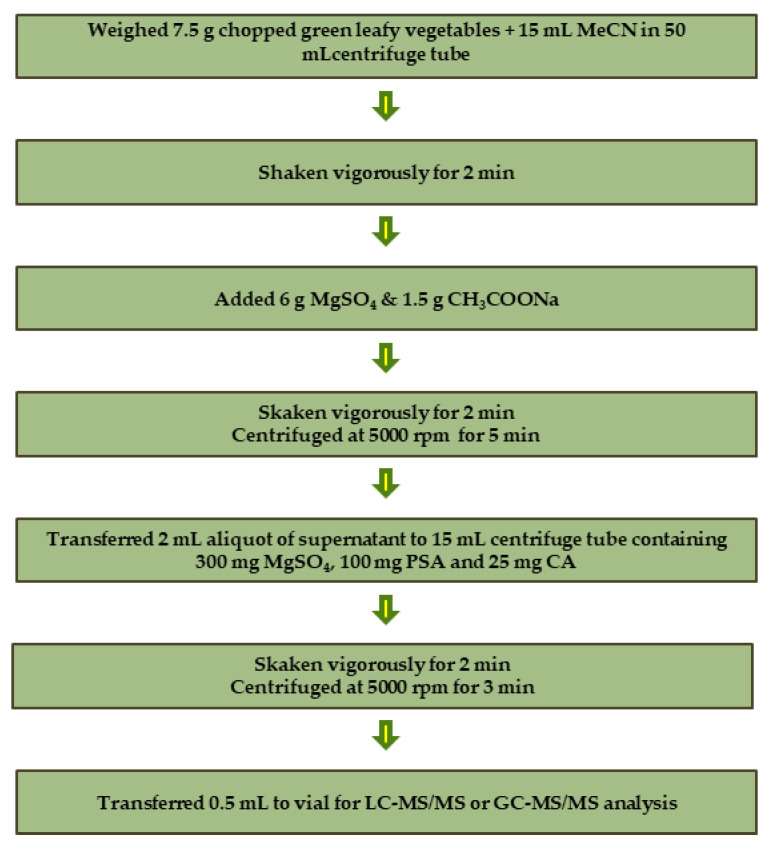
Schematic flow diagram of the QuEChERS sample preparation.

**Figure 2 foods-12-01034-f002:**
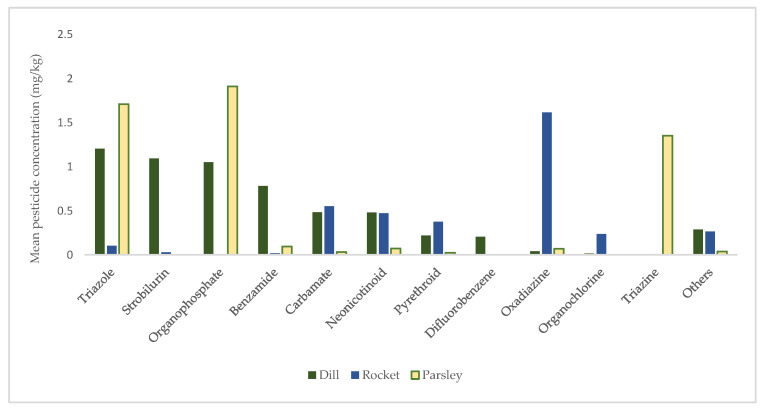
Main class of pesticides detected in green leafy vegetables.

**Figure 3 foods-12-01034-f003:**
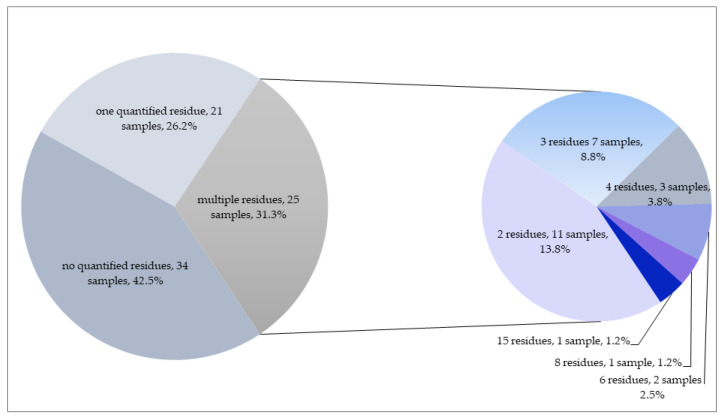
Number of quantified residues in dill samples.

**Figure 4 foods-12-01034-f004:**
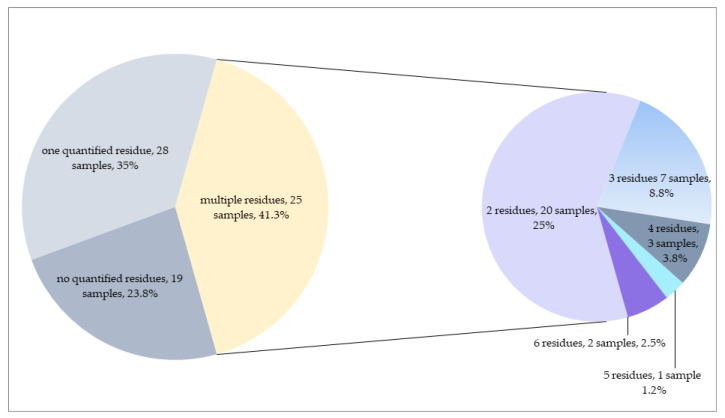
Number of quantified residues in rocket samples.

**Figure 5 foods-12-01034-f005:**
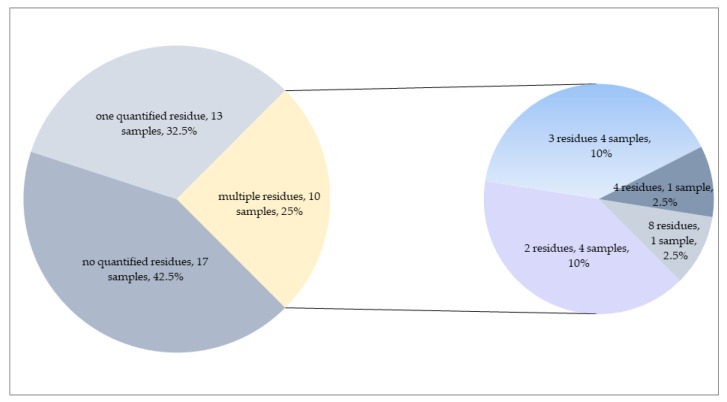
Number of quantified residues in parsley samples.

**Table 1 foods-12-01034-t001:** Analytical conditions for LC.

Parameter	Chromatographic Conditions		
Column:	Accucore Vanquish C-18 (2.1 × 100 mm, 1.5 µm particle size, ThermoFisher Scientific)		
Column temperature:	35 °C		
Injection volume:	10 µL		
Flow rate:	0.6 mL min^−1^		
Mobile phase A:	Water-methanol (98:2, *v*/*v*) with 0.1% formic acid and 5 mM ammonium formate		
Mobile phase B:	Methanol		
LC gradient program:	Gradient time (min)	A%	B%
	0.0	80	20
	0.2	80	20
	1.5	30	70
	6.0	5	95
	7.5	5	95
	7.6	80	20
	10.0	80	20

**Table 2 foods-12-01034-t002:** The presence and quantification of pesticide residues in dill.

Pesticide	Type of Residue	EU MRL (mg kg^−1^)	% of Samples <LOQ	% of Samples between LOQ-MRL	% of Samples >MRL	Range (mg kg^−1^)	
						Min.–Max.	Mean
Acetamiprid	IN ^a^	0.05	96.3	1.2	2.5	0.021–1.179	0.430
Azoxystrobin	FU ^b^	0.3	92.5	2.5	5	0.013–2.744	1.092
Chlorpyrifos *	IN/AC ^c^	0.01	95.0	-	5	0.010–0.023	0.014
Chlorpyrifos-methyl *	IN/AC	0.01	98.8	-	1.2	0.867	0.867
Cyflufenamid	FU	0.05	96.3	2.5	1.2	0.018–0.569	0.204
Deltamethrin	IN	0.1	97.5	1.2	1.2	0.044–0.305	0.175
Difenoconazole	FU	0.3	92.5	6.2	1.2	0.015–0.525	0.188
Dimethomorph	FU	30.0	98.8	1.2	-	0.017	0.017
Diuron	HB ^d^	0.05	96.3	3.7	-	0.010–0.020	0.014
Ethofumesate	HB	0.06	98.8	1.2	-	0.010	0.010
Fipronil *	IN	0.005	98.8	1.2	-	0.013	0.013
Fludioxonil	FU	0.05	98.8	1.2	-	0.018	0.018
Fluopyram	FU	6.0	88.8	11.2	-	0.016–3.915	0.780
Hexaconazole *	FU	0.05	98.8	1.2	-	0.012	0.012
Indoxacarb *	IN	0.05	96.3	2.5	1.2	0.023–0.068	0.039
Lambda cyhalothrin	IN	0.3	98.8	1.2	-	0.042	0.042
Malathion	FU	0.02	96.3	-	3.7	0.030–0.374	0.170
Metalaxyl-M	FU	0.05	93.8	5.0	1.2	0.015–0.195	0.060
Methiocarb *	IN	0.01	98.8	-	1.2	0.155	0.155
Methoxyfenozide	IN	0.05	98.8	-	1.2	0.057	0.057
Metribuzin	HB	0.01	97.5	-	2,5	0.026–0.030	0.028
Metolachlor *	HB	0.05	98.8	1.2	-	0.015	0.015
Penconazole	FU	0.05	95.0	1.2	3.8	0.030–0.590	0.188
Pendimethalin	HB	0.05	77.5	17.5	5	0.016–0.077	0.037
Pirimicarb	IN	5.0	97.5	2.5	-	0.011–0.615	0.313
Pyrimethanil	FU	0.05	98.8	1.2	-	0.033	0.033
Tebuconazole	FU	1.5	86.3	12.5	1.2	0.010–2.077	0.523
Thiacloprid *	IN	0.08	96.3	3.7	-	0.018–0.063	0.034
Thiamethoxam *	IN	0.05	97.5	2.5	-	0.012–0.017	0.015
Triadimenol *	FU	0.05	96.3	1.2	2.5	0.023–0.156	0.086
Tri-allate	HB	0.1	93.8	6.2	-	0.010–0.022	0.014
Triflumizole *	FU	0.1	96.3	-	3.7	0.017–0.583	0.207

^a^ IN: insecticide. ^b^ FU: fungicide. ^c^ AC: acaricide. ^d^ HB: herbicide. * Not approved in the EU.

**Table 3 foods-12-01034-t003:** The presence and quantification of pesticide residues in rocket.

Pesticide	Type of Residue	EU MRL (mg kg^−1^)	% of Samples <LOQ	% of Samples between LOQ-MRL	% of Samples >MRL	Range (mg kg^−1^)	
						Min.–Max.	Mean
Acetamiprid	IN ^a^	3.0	70.0	30.0	-	0.010–0.471	0.180
Azoxystrobin	FU ^b^	15.0	93.7	6.3	-	0.011–0.021	0.016
Bifenthrin *	IN	0.01	97.5	-	2.5	0.012–0.013	0.013
Clothianidin *	IN/AC ^c^	0.01	97.5	-	2.5	0.032–0.033	0.032
Cypermethrin	IN	2.0	93.7	6.3	-	0.028–0.921	0.313
Deltamethrin	IN	2.0	90.0	10.0	-	0.013–0.070	0.028
Dimethomorph	FU	10.0	98.8	1.2	-	0.089	0.089
Dinocap *	FU	0.02	95.0	-	5.0	0.032–0.058	0.041
Diuron *	HB ^d^	0.01	61.3	-	38.7	0.010–0.042	0.036
Fluopyram	FU	20.0	96.3	3.7	-	0.015–0.022	0.018
Hexythiazox	AC	0.01	98.8	-	1.2	0.018	0.018
Imidacloprid *	IN	0.01	98.8	-	1.2	0.058	0.058
Indoxacarb *	IN	2.0	96.3	2.5	1.2	0.728–2.615	1.615
Lambda cyhalothrin	IN	0.7	93.7	6.3	-	0.014–0.029	0.020
Metalaxyl-M	FU	3.0	92.5	7.5	-	0.017–0.297	0.081
Oxamyl	IN/NE ^e^	0.01	98.8	-	1.2	0.040	0.040
Penconazole	FU	0.01	98.8	-	1.2	0.013	0.013
Pirimicarb	IN	15.0	98.8	1.2	-	0.510	0.510
Pyridaben	IN/AC	0.01	96.3	-	3.7	0.013–0.681	0.236
Tebuconazole	FU	0.5	98.8	1.2	-	0.028	0.028
Thiamethoxam *	IN	0.01	91.3	-	8.7	0.019–0.614	0.202
Triadimenol *	FU	0.01	98.7	-	1.2	0.061	0.061
Trifloxystrobin	FU	15.0	98.8	1.2	-	0.012	0.012

^a^ IN: insecticide. ^b^ FU: fungicide. ^c^ AC: acaricide. ^d^ HB: herbicide. ^e^ NE: nematicide. * Not approved in the EU.

**Table 4 foods-12-01034-t004:** The presence and quantification of pesticide residues in parsley.

Pesticide	Type of Residue	EU MRL (mg kg^−1^)	% of Samples <LOQ	% of Samples between LOQ-MRL	% of Samples >MRL	Range (mg kg^−1^)	
						Min.–Max.	Mean
Acetamiprid	IN ^a^	3.0	87.5	12.5	-	0.014–0.074	0.044
Chlorpyrifos-methyl *	IN/AC ^b^	0.01	97.5	-	2.5	0.899	0.899
Cypermethrin	IN	2.0	92.5	7.5	-	0.019–0.029	0.024
Ethoprophos *	IN/AC	0.02	97.5	2.5	-	0.018	0.018
Fluopyram	FU ^c^	6.0	95.0	5.0	-	0.027–0.158	0.093
Hexaconazole *	FU	0.02	97.5	-	2.5	1.501	1.501
Imidacloprid *	IN	0.05	90.0	10.0	-	0.012–0.040	0.027
Indoxacarb *	IN	2.0	97.5	2.5	-	0.068	0.068
Malathion	FU	0.02	97.5	-	2.5	0.991	0.991
Methiocarb *	IN	0.06	97.5	2.5	-	0.030	0.030
Pendimethalin	HB ^d^	2.0	95.0	5.0	-	0.013–0.020	0.016
Pymetrozine *	IN	0.05	47.5	2.5	50.0	0.016–2.437	1.351
Pyrimethanil	FU	20.0	97.5	2.5	-	0.020	0.020
Tebuconazole	FU	2.0	97.5	2.5	-	0.207	0.207

^a^ IN: insecticide. ^b^ AC: acaricide. ^c^ FU: fungicide. ^d^ HB: herbicide. * Not approved in the EU.

## Data Availability

Data is contained within the article.

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
