# Peer review of "Quantification of 363 Pesticides in Leafy Vegetables (Dill, Rocket and Parsley) in the Turkey Market by Using QuEChERS with LC-MS/MS and GC-MS/MS"

_foods, 2023, doi:10.3390/foods12051034_

Round 1

Reviewer 1 Report

1. The discussion part lacks information on the pesticide contamination of leafy vegetables across the world. 

2. There need for remedial measures to be taken up especially acetamiprid and malathion detected in the sample.

3. Dissipation studies are needed for wholesome conclusions.

4. Compulsory conduction of various treatments for leafy vegetables and then profiling of pesticides is to be shown in the paper. This work would add certain uniqueness and note worthy. Only screening and determination would not solve the problem and any benefit to the consumers.

5. More recent References have to be quoted and the number of references is scanty. 

6. It should have been a comparative study between Corum and other cities in Turkey to give a picture.

Author Response

Reviewer #1 Report

1.The discussion part lacks information on the pesticide contamination of leafy vegetables across the world. 

  • The pesticide contamination of leafy vegetables across the world have now been more described in the discussion section.
  1. There need for remedial measures to be taken up especially acetamiprid and malathion detected in the sample.
  • The sentences “For this reason, pesticide residues need to be remediated to prevent adverse effects on humans and animals. The use of microorganisms such as microalgae is the most effective, eco-friendly and economical method in bioremediation to detoxify or convert hazardous compounds into non-toxic compounds” have been inserted into the text.
  1. Dissipation studies are needed for wholesome conclusions.
  • Unfortunately, this work does not contain dissipation studies.
  1. Compulsory conduction of various treatments for leafy vegetables and then profiling of pesticides is to be shown in the paper. This work would add certain uniqueness and note worthy. Only screening and determination would not solve the problem and any benefit to the consumers.
  • Of course, the removal of pesticide residues from fresh produce is important to reduce pesticide exposure to humans. Different techniques aim at either degrading or removal of pesticides from the food. However, there are only a few data on the levels of pesticide residues in green leafy vegetables from Turkey. Thus, the manuscript mainly concentrated on the quantitation of multiple residues in ready-to-eat green leafy vegetables. This study underlines the importance of periodic monitoring of pesticide residues in green leafy vegetables from Turkey.
  1. More recent References have to be quoted and the number of references is scanty. 
  • Thank you for your advice. More recent references are now cited and the number of references is increased from 15 to 41.
  1. It should have been a comparative study between Corum and other cities in Turkey to give a picture.
  • The samples were collected from supermarkets, groceries and bazaars in Corum, Turkey. The samples were not taken from other cities. However, green leafy vegetables sold throughout the country primarily originated from the Mediterranean region of Turkey, particularly in Hatay province, followed by Thrace and Aegean regions. In this concept, a comparative study was not studied.

Reviewer 2 Report

Screening and Quantitation of Multiclass Pesticide Residues in Leafy Vegetables using LC-MS/MS and GC-MS/MS

Overall

Analysis and quantification of pesticides in green leafy vegetables has been carried out by many previous studies. This study performed the analysis of pesticides in leafy vegetables on LC/MS/MS and GC/MS/MS equipment. The study collected 200 samples of leafy vegetables (including 80 dill, 80 rocket and 40 parsley) at green grocer shops, markets and bazaar in Corum province, Turkey. The study used a sample extraction method by QuEChERS and analyzed on LC/MS/MS ThermoFisher Scientific Q-Exactive (a high resolution) and GC/MS/MS GC-MS TQ8050, Shimadzu equipment. The results showed that 35% of samples did not contain pesticide residues, 65% of samples detected 43 different pesticide residues. In which, rocket samples were detected the most (61 samples/80 samples, 76.3%), then dill (46 samples/80 samples, 57.5%) and parsley (23 samples/40 samples 57.5%). 46% of the samples exceeded the regulation of EU MRLs.

Overall, this is a pretty simple study, not much of a novelty. The manuscript is not well written, there is not much argument and discussion. The manuscript is presented more like a food safety inspection report than a scientific report.

Questions:

1.     Why does this study only collect 200 samples for analysis and each type only takes 40-80 samples? Why research only 3 vegetables dill, roket and parsley? While some other vegetables such as lettuce, spinach and some other vegetables are more commonly used in Europe have not been studied?

2.     There are many analytical methods, why choose the QuEChERS method for this study?

3.     For each vegetable, only a few pesticides are used for agriculture. Please explain why this study analyzed many pesticides present in vegetables? Where do the sources of these pesticides come from?

Specific comments:

I would like to comment from top to bottom of manuscript

1.     Title of manuscript: has not covered all the presented contents. The title has not shown the novelty of the research manuscript. The word “Screening” is inappropriate for this title of manuscript, because the study did not use the screening method. The word “Multiclass Pesticide Residues” is appropriate, but the research manuscript contents are not categorized in the multiclass groups of pesticides.

2.     Abstract: has generally summarized the manuscript results, but because the research is not novel and has no interest, it is necessary to revise the entire manuscript to be able then to present again the abstract.

3.     Introduction: has not presented the necessity of the study issues. Many presented contents have not been related to the research, for example, the situation of used pesticide in the world such as the United States, Brazil, and China. This section should directly describe Turkey's agriculture as the vegetable production and consumption. How were the studied vegetables consumed? What pesticides are used in agriculture?

This section should also cover the analytical method of pesticides, mainly to clarify why to use QuEChERS, LC/MS/MS, GC/MS/MS. Corum province's green vegetable distribution system should be described.

4.     The section of materials and methods: has not clearly been presented in detail. The collected samples did not present which markets were purchased. It is necessary to list the markets in the supplementary table. No presentation of duplicate samples has been found in the manuscript. It is necessary to specify which source the vegetable samples supplied to the markets? How much are the vegetable suppliers for all the markets in Corum province? What is their agricultural farming? How to use pesticides? The quantification of large pesticide amounts in vegetables could negatively impact the prestige of local agricultural enterprises in Corum province. Therefore, it should be careful before publishing the manuscript.

The sample preparation procedure requires homogenization using a blender or homogenizer. Figure 1 is described by the sentence in lines from 105-107: “After centrifugation, the supernatant (0.5 mL) was diluted with water (1:3 ratios, v/v), and the diluted extract was transferred into the LC vial for chromatographic analysis” is not reasonable, because the final dilution solvents are different between LC and GC injection. A more detailed description of sample preparation is required. There are many different types of QuEChERS, which one should be described in the study? The study did not use internal standards and surrogates, so it is difficult to evaluate the accuracy and recovery of the method.

LC/MS/MS and GC/MS/MS analysis are also not well presented. It should be described the MS conditions, what are the product and precursor ions monitored? On the other hand, the manuscript does not present the different collision energy conditions. The study also did not describe calibration curves. This presentation can be included in the supplementary data. It is not clear if the study uses the library of the Q-Exactive LC/MS/MS instrument?

5.     It is necessary to separate table S1 into 2 tables for LC/MS/MS and GC/MS/MS. It is not clear why the study selected concentrations of 0.01 and 0.05 mg.kg-1 to evaluate repeatability and reproducibility. Please describe in more detail how the within-laboratory reproducibility was carried out? The results of section 3.1. Validation data needs to be shown graphically.

The results of pesticide residues in leafy vegetables are presented in three separated categories according to each vegetable: 3.2.1. Residue data on dill, 3.2.2. Residue data on rocket, 3.2.3. Residue data on parsley. These sections have been presented with almost the same content layout, differing only in the result data of pesticides in vegetables. The presentation of the result was not very creative. It is necessary to divide into different groups of pesticides and express them in a chart in order to make the reader easier evaluate the pesticide concentrations in vegetables. The result will be more interesting if presented well. The result section should compare the detected pesticides in vegetables and discuss the origin sources of those pesticides.

6.     The English should be all revised to make it better. The manuscript also has some typos such as "QuChERS" in line 13 that needs to be corrected as "QuEChERS". Please check carefully for spelling errors.

7.     The article should cite more references.

Conclusion:

In general, this manuscript is quite simply presented, the draft of the article is not well written, there is not much discussion and argument about the research results. The manuscript does not have much novelty. The number of pesticides studied up to 363 compounds is also quite a large data, it needs to better exploit the research results. Therefore, the article needs to be carefully revised a lot of content.

Author Response

Reviewer #2 Report

Overall

Analysis and quantification of pesticides in green leafy vegetables has been carried out by many previous studies. This study performed the analysis of pesticides in leafy vegetables on LC/MS/MS and GC/MS/MS equipment. The study collected 200 samples of leafy vegetables (including 80 dill, 80 rocket and 40 parsley) at green grocer shops, markets and bazaar in Corum province, Turkey. The study used a sample extraction method by QuEChERS and analyzed on LC/MS/MS ThermoFisher Scientific Q-Exactive (a high resolution) and GC/MS/MS GC-MS TQ8050, Shimadzu equipment. The results showed that 35% of samples did not contain pesticide residues, 65% of samples detected 43 different pesticide residues. In which, rocket samples were detected the most (61 samples/80 samples, 76.3%), then dill (46 samples/80 samples, 57.5%) and parsley (23 samples/40 samples 57.5%). 46% of the samples exceeded the regulation of EU MRLs.

Overall, this is a pretty simple study, not much of a novelty. The manuscript is not well written, there is not much argument and discussion. The manuscript is presented more like a food safety inspection report than a scientific report.

  • The manuscript has been improved.

Questions:

  1. Why does this study only collect 200 samples for analysis and each type only takes 40-80 samples? Why research only 3 vegetables dill, roket and parsley? While some other vegetables such as lettuce, spinach and some other vegetables are more commonly used in Europe have not been studied?

  • The samples were analyzed for over 360 pesticides. The number of samples more than 200 could be better. However, it can not be said that the number of samples (n=200) is insufficient for a survey study. The quantification of pesticides in other green leafy vegetables such as lettuce and spinach has been carried out more commonly. However, little data on the presence of pesticides in targeted leafy vegetables. Thus, this study mainly concentrated on the levels of 363 pesticides in dill, rocket and parsley.

  1. There are many analytical methods, why choose the QuEChERS method for this study?

A variety of sample preparation methods have been used for extracting pesticide residues in various food products. Traditional sample preparation procedures include solid-phase extraction, solid-phase microextraction, accelerated solvent extraction, supercritical fluid extraction, matrix solid-phase dispersion, microwave-assisted extraction and gel permeation chromatography. However, the majority of these techniques are rather time-consuming, labor-intensive, complicated, expensive and produce considerable quantities of waste. The QuEChERS (quick, easy, cheap, effective, rugged, and safe) is the most popular current sample preparation method in pesticide analysis. Due to its simplicity, speed, low-cost and high-throughput, and minimal solvent requirement, the QuEChERS method has been chosen.

  1. For each vegetable, only a few pesticides are used for agriculture. Please explain why this study analyzed many pesticides present in vegetables? Where do the sources of these pesticides come from?

  • While acetamiprid, azoxystrobin, deltamethrin and quizalofop-p are the most common pesticides used by farmers in the cultivation of conventional green leafy vegetables in Turkey, many other residues can also be detected in the product. The presence of many different pesticides in leafy vegetables could be not only through the direct application of active substances to leafy vegetables but also through other sources such as water resources and the environment. Pesticides can be transported by factors such as wind erosion, evaporation, and attachment to soil particles. The main factors affecting the transport are water solubility (hydrophobic-hydrophilic), volatility, adsorption in water or soil, bioaccumulation amount in soil and biota, and chemical-microbial degradation properties.

Specific comments:

I would like to comment from top to bottom of manuscript

  1. Title of manuscript: has not covered all the presented contents. The title has not shown the novelty of the research manuscript. The word “Screening” is inappropriate for this title of manuscript, because the study did not use the screening method. The word “Multiclass Pesticide Residues” is appropriate, but the research manuscript contents are not categorized in the multiclass groups of pesticides.

  • Thank you for your comments. The title of the manuscript has been changed to “Quantitation of more than 360 pesticide residues in leafy vegetables using LC-MS/MS and GC-MS/MS”.

  1. Abstract: has generally summarized the manuscript results, but because the research is not novel and has no interest, it is necessary to revise the entire manuscript to be able then to present again the abstract.

  • Thank you for your comments. This manuscript mainly focused on the quantitation of multiple residues in ready-to-eat green leafy vegetables and the results bring us some information about pesticide residues in green leafy vegetables consumed in Turkey. The current abstract gives us information about the purpose of the research, methods used in the analysis and the principal results. For that reason, we think that the abstract seems to be appropriate.

  1. Introduction: has not presented the necessity of the study issues. Many presented contents have not been related to the research, for example, the situation of used pesticide in the world such as the United States, Brazil, and China. This section should directly describe Turkey's agriculture as the vegetable production and consumption. How were the studied vegetables consumed? What pesticides are used in agriculture?

This section should also cover the analytical method of pesticides, mainly to clarify why to use QuEChERS, LC/MS/MS, GC/MS/MS. Corum province's green vegetable distribution system should be described.

  • Thank you for your comments. The introduction section has been improved as suggested by reviewer 2.

  1. The section of materials and methods: has not clearly been presented in detail. The collected samples did not present which markets were purchased. It is necessary to list the markets in the supplementary table. No presentation of duplicate samples has been found in the manuscript. It is necessary to specify which source the vegetable samples supplied to the markets? How much are the vegetable suppliers for all the markets in Corum province? What is their agricultural farming? How to use pesticides? The quantification of large pesticide amounts in vegetables could negatively impact the prestige of local agricultural enterprises in Corum province. Therefore, it should be careful before publishing the manuscript.

The sample preparation procedure requires homogenization using a blender or homogenizer. Figure 1 is described by the sentence in lines from 105-107: “After centrifugation, the supernatant (0.5 mL) was diluted with water (1:3 ratios, v/v), and the diluted extract was transferred into the LC vial for chromatographic analysis” is not reasonable, because the final dilution solvents are different between LC and GC injection. A more detailed description of sample preparation is required. There are many different types of QuEChERS, which one should be described in the study? The study did not use internal standards and surrogates, so it is difficult to evaluate the accuracy and recovery of the method.

LC/MS/MS and GC/MS/MS analysis are also not well presented. It should be described the MS conditions, what are the product and precursor ions monitored? On the other hand, the manuscript does not present the different collision energy conditions. The study also did not describe calibration curves. This presentation can be included in the supplementary data. It is not clear if the study uses the library of the Q-Exactive LC/MS/MS instrument?

  • The samples were taken from seven different supermarkets, eight different groceries and five different bazaars in Corum province. While each supermarket is used its own distribution channels, six green leafy vegetable suppliers are available in Corum for all groceries and bazaars.
  • The unwashed samples were chopped, homogenised with a home food processor (Phillips HR 7770, Royal Philips Electronics NV, Turkey) and stored in a plastic container in a refrigerator (4–8oC) until analysis. Pesticide residues were extracted from green leafy vegetables using the QuEChERS extraction method according to AOAC official method (AOAC, 2007), with minor modifications.
  • The study has been described the calibration studies. The LC-MS/MS and GC-MS/MS conditions are now more described in the manuscript.

  1. It is necessary to separate table S1 into 2 tables for LC/MS/MS and GC/MS/MS. It is not clear why the study selected concentrations of 0.01 and 0.05 mg.kg-1 to evaluate repeatability and reproducibility. Please describe in more detail how the within-laboratory reproducibility was carried out? The results of section 3.1. Validation data needs to be shown graphically.

The results of pesticide residues in leafy vegetables are presented in three separated categories according to each vegetable: 3.2.1. Residue data on dill, 3.2.2. Residue data on rocket, 3.2.3. Residue data on parsley. These sections have been presented with almost the same content layout, differing only in the result data of pesticides in vegetables. The presentation of the result was not very creative. It is necessary to divide into different groups of pesticides and express them in a chart in order to make the reader easier evaluate the pesticide concentrations in vegetables. The result will be more interesting if presented well. The result section should compare the detected pesticides in vegetables and discuss the origin sources of those pesticides.

  • The supplementary material is divided into two parts as S1 and S2 for LC-MS/MS and GC-MS/MS, respectively. To evaluate the recovery and precision of the method, the targeted analytes were spiked into blank leafy vegetable matrices at the targeted limit of quantifications of 0.01 mg kg-1, and 5 x targeted LOQ (0.05 mg kg-1), each at five replicates. The repeatability was determined on the same day by one operator (n= 5), while within-laboratory reproducibility was assessed on five consecutive days by two different operators (n= 10). The validation data are summarized in Tables S1 and S2.
  • The discussion section has been improved.

  1. The English should be all revised to make it better. The manuscript also has some typos such as "QuChERS" in line 13 that needs to be corrected as "QuEChERS". Please check carefully for spelling errors.

  • The WHOLE manuscript has been revised in-depth, and some typographical and grammatical errors have been removed.

  1. The article should cite more references.

  • Thank you for your advice. More recent references are now cited and the number of references is increased from 15 to 41.

Conclusion:

In general, this manuscript is quite simply presented, the draft of the article is not well written, there is not much discussion and argument about the research results. The manuscript does not have much novelty. The number of pesticides studied up to 363 compounds is also quite a large data, it needs to better exploit the research results. Therefore, the article needs to be carefully revised a lot of content.

  • The manuscript has been revised and improved.

Round 2

Reviewer 2 Report

2nd Review: Screening and Quantitation of Multiclass Pesticide Residues in Leafy Vegetables using LC-MS/MS and GC-MS/MS

The manuscript has been generally revised and adjusted according to the comments of the reviewer. However, I still have some suggestions to make a better article.

1.     The title of the manuscript has been revised into: “Quantitation of more than 360 Pesticide Residues in Leafy Vegetables using LC-MS/MS and GC-MS/MS". However, I still want the title to be more detailed about the research problem and to distinguish it from other research articles. Therefore, the names of the three vegetables: dill, rocket and parsley should be added to the title and the QuEChERS treatment sample method should be also added. I suggest the example title: “Quantification of 363 pesticides in leafy vegetables (dill, rocket and parsley) in the Turkey market by using QuEChERS LC-MS/MS and GC-MS/MS”. The word “Quantification” should be used instead of “Quantitation”.

2.     The introduction section has been appropriately revised. I would like the manuscript to cite one more reference: “Nam Vu Duc, Trung Nguyen Quang, Thuy Le Minh, Xuyen Nguyen Thi, Tri Manh Tran, Hai Anh Vu, Lan Anh Nguyen, Tien Doan Duy, Bui Van Hoi, Cam Tu Vu, Dung Le Van, Lan Anh Phung Thi, Hong An Vu Thi, Dinh Binh Chu, 2019, Multiresidue Pesticides Analysis of Vegetables in Vietnam by Ultrahigh-Performance Liquid Chromatography in Combination with High-Resolution Mass Spectrometry (UPLC-Orbitrap MS ), Journal of Analytical Methods in Chemistry, https://doi.org/10.1155/2019/3489634. This reference is quite similar to this study on using QuEChERS, LC-Orbitrap and also on vegetables.

3.     The research results have not been evaluated for groups of pesticides. It is necessary to divide the detected pesticides into different groups to assess the presence of groups. It is recommended to present the column charts of detected pesticide concentration groups to make the results more vivid. The pesticides were hypothesized from other sources such as water resources and the environment. However, it should be presented in more detail by a number of references. For example, is there data on the use of these pesticides and their occurrence in surface water in Turkey? This conclusion is quite important because it will affect the country's vegetable farming industry.

The manuscript did not compare the concentrations of pesticides and pesticide groups detected on 3 vegetables: dill, rocket and parsley. The comparison of the amount present will show which vegetables absorb more pesticides. Can be added as a separate section of comparison.

The phrases from line 345-371 are presented as an introduction, not the results of this study.

In general, the manuscript has better been revised, but it is necessary to further discuss the results, especially about the groups of pesticides and compare the occurrence of pesticides/pesticide groups in vegetables.

Author Response

Manuscript ID: foods-2197622 (2nd review)

 Reviewer #2 Report

The manuscript has been generally revised and adjusted according to the comments of the reviewer. However, I still have some suggestions to make a better article.

1.The title of the manuscript has been revised into: “Quantitation of more than 360 Pesticide Residues in Leafy Vegetables using LC-MS/MS and GC-MS/MS". However, I still want the title to be more detailed about the research problem and to distinguish it from other research articles. Therefore, the names of the three vegetables: dill, rocket and parsley should be added to the title and the QuEChERS treatment sample method should be also added. I suggest the example title: “Quantification of 363 pesticides in leafy vegetables (dill, rocket and parsley) in the Turkey market by using QuEChERS LC-MS/MS and GC-MS/MS”. The word “Quantification” should be used instead of “Quantitation”.

  • Thank you for your comments. The title of the manuscript has been changed to “Quantification of 363 pesticides in leafy vegetables (dill, rocket and parsley) in the Turkey market by using QuEChERS with LC-MS/MS and GC-MS/MS”.

2.The introduction section has been appropriately revised. I would like the manuscript to cite one more reference: “Nam Vu Duc, Trung Nguyen Quang, Thuy Le Minh, Xuyen Nguyen Thi, Tri Manh Tran, Hai Anh Vu, Lan Anh Nguyen, Tien Doan Duy, Bui Van Hoi, Cam Tu Vu, Dung Le Van, Lan Anh Phung Thi, Hong An Vu Thi, Dinh Binh Chu, 2019, Multiresidue Pesticides Analysis of Vegetables in Vietnam by Ultrahigh-Performance Liquid Chromatography in Combination with High-Resolution Mass Spectrometry (UPLC-Orbitrap MS), Journal of Analytical Methods in Chemistry, https://doi.org/10.1155/2019/3489634. This reference is quite similar to this study on using QuEChERS, LC-Orbitrap and also on vegetables.

  • The reference “Vu Duc, N.; Nguyen-Quang, T.; Le-Minh, T.; Nguyen-Thi, X.; Tran, T.M.; Vu, H.A.; Nguyen, L.-A.; Doan-Duy, T.; Hoi, B.V.; Vu, C.-T.; Le-Van, D.; Phung-Thi, L.-A.; Vu-Thi, H.-A.; Chu, D.B. Multiresidue pesticides analysis of vegetables in Vietnam by ultrahigh-performance liquid chromatography in combination with high-resolution mass spectrometry (UPLC-orbitrap MS). Anal. Methods Chem. 2019, 3489634” has been cited.

3.The research results have not been evaluated for groups of pesticides. It is necessary to divide the detected pesticides into different groups to assess the presence of groups. It is recommended to present the column charts of detected pesticide concentration groups to make the results more vivid. The pesticides were hypothesized from other sources such as water resources and the environment. However, it should be presented in more detail by a number of references. For example, is there data on the use of these pesticides and their occurrence in surface water in Turkey? This conclusion is quite important because it will affect the country's vegetable farming industry. The manuscript did not compare the concentrations of pesticides and pesticide groups detected on 3 vegetables: dill, rocket and parsley. The comparison of the amount present will show which vegetables absorb more pesticides. Can be added as a separate section of comparison.

The phrases from line 345-371 are presented as an introduction, not the results of this study. In general, the manuscript has better been revised, but it is necessary to further discuss the results, especially about the groups of pesticides and compare the occurrence of pesticides/pesticide groups in vegetables.

  • Thank you for your comments. The groups of pesticides based on chemical class detected in green leafy vegetables (dill, rocket and parsley) are now shown in Figure 2. The research results have been evaluated for groups of pesticides.
  • Several studies have been performed to detect pesticide contamination in surface water and sediments in Turkey. The results of studies on the occurrence of pesticide residues in the surface water of Turkey have been described in the manuscript.
  • The manuscript has been revised and improved.